# Therapeutic Approach to Post-Transplant Recurrence of Hepatocellular Carcinoma: Certainties and Open Issues

**DOI:** 10.3390/cancers15235593

**Published:** 2023-11-26

**Authors:** Giuseppe Marrone, Maria Sandrina Leone, Marco Biolato, Antonio Liguori, Giuseppe Bianco, Gabriele Spoletini, Antonio Gasbarrini, Luca Miele, Maurizio Pompili

**Affiliations:** Medical and Surgical Sciences Department, Policlinico Universitario A. Gemelli-IRCCS, Università Cattolica del Sacro Cuore, 00168 Rome, Italy

**Keywords:** liver transplantation, HCC recurrence, vascular invasion, mTOR inhibitors, checkpoint inhibitors

## Abstract

**Simple Summary:**

Hepatocellular carcinoma (HCC) recurrence after liver transplantation (LT) results in a relevant worsening of the prognosis of transplanted subjects. Pre-transplant HCC characteristics are the main determinants of the risk of recurrence and various tools exist to estimate recurrence risk. Once recurrence has occurred, the pattern of recurrence greatly influences whether the patient is a candidate for curative treatment and consequently significantly impacts prognosis. We reviewed different predictive models of post-transplant recurrence, the role of immunosuppression, and the efficacy and feasibility of various therapeutic approaches.

**Abstract:**

Hepatocellular carcinoma (HCC) is a growing indication for liver transplantation (LT). Careful candidate selection is a prerequisite to keep post-LT recurrence rates within acceptable percentages. In the pre-LT period, various types of locoregional treatments and/or systemic therapies can be used for bridging or downstaging purposes. In this context, one of the factors limiting the possibility of treatment is the degree of functional liver impairment. In the LT subject, no widely accepted indications are available to guide treatment of disease recurrence and heterogeneity exists between transplant centers. Improved liver function post LT makes multiple therapeutic strategies theoretically feasible, but patient management is complicated by the need to adjust immunosuppressive therapy and to assess potential toxicities and drug–drug interactions. Finally, there is controversy and uncertainty about the use of recently introduced immunotherapeutic drugs, mainly due to the risk of organ rejection. In this paper, we will review the most recent available literature on the management of post-transplant HCC recurrence, discussing evidence and controversies.

## 1. Introduction

Liver transplantation (LT) is the ideal curative treatment for hepatocellular carcinoma (HCC) as it can remove not only liver cancer but also resolve liver cirrhosis, which is the most important cause of hepatocarcinogenesis [1].

In patients with significantly impaired liver function, LT can be the only viable curative approach for HCC.

The prerequisite for the use of LT as a therapy for HCC is that the neoplasm must be localized exclusively in the liver, as the presence of extrahepatic localization would be accompanied by a universal post-LT HCC recurrence.

Pre-LT imaging techniques can identify macroscopic metastases but do not guarantee the exclusion of microscopic metastases. To prevent post-transplant recurrence, various selection criteria for LT candidates have been proposed over the years with the intent to reduce the risk of misrecognized micro-metastases. The use of such criteria allows for the recurrence rate to be maintained within acceptable rates.

In this narrative review, we will review the main scientific evidence regarding selection criteria for transplant candidates, risk factors for recurrence, and treatment of HCC recurrence. In conducting the review, a literature search in the PubMed database was performed by two of the authors (GM and MSL) independently. Papers published up to 31 August 2023 and concerning selection criteria for transplantation, risk factors for recurrence, post-transplantation surveillance, and recurrence treatments were considered. Papers were selected based on title and abstract and, if deemed relevant based on the authors’ arbitrary judgment for the conduct of the review, the full text was retrieved and fully evaluated.

## 2. Selection Criteria for LT

In 1996, Mazzaferro et al. first established morphological criteria for successful transplantation for HCC, proposing what later became known as the “Milan criteria”. Transplantable subjects must present with a single lesion with a maximum diameter ≤ 5 cm or up to three separate lesions no larger than 3 cm, along with no evidence of gross vascular invasion and no regional nodal or distant metastases. Patients who satisfied these criteria showed a 4-year survival rate of 75%, similar to expected survival rates for patients undergoing transplantation for cirrhosis without HCC. In the original Mazzaferro study, only 8% of enrolled patients experienced HCC recurrence within 4 years [2,3].

Over the years, less restrictive criteria have been proposed with comparable results to the Milan criteria. In 2001, the University of California, San Francisco (UCSF) proposed new extended criteria which are fulfilled if the patient has a single nodule up to 6.5 cm in maximum diameter or three nodules with a total diameter of up to 8 cm. These criteria led to a favorable 5-year patient survival rate of 75.2% and a post-LT recurrence rate of 11.4% [4].

In 2009, Mazzaferro et al. proposed the so-called “up-to-seven” criteria. Using these criteria, subjects are considered for OLT if the sum of the number of nodules and the diameter (in cm) of the largest nodule does not exceed seven [5] (Table 1).

Several studies have evaluated the reliability of the up-to-seven criteria as inclusion criteria, showing post-LT survival rates roughly overlapping with patients meeting the Milan criteria [6,7].

All the mentioned criteria are based on the morphological characteristics of the HCC and are built on the pre-LT accuracy of liver imaging techniques such as contrast-enhanced computed tomography or magnetic resonance imaging, which can be affected by heterogeneity and suboptimal sensitivity [8,9,10].

For patients with a tumor stage beyond morphologic criteria, the possibility of tumor stage reduction in order to fall within morphologic criteria (i.e., “Downstaging”) using various treatments (including either surgery, ablation, chemoembolization, radioembolization, or systemic treatments) has been explored with good results [11,12,13]. Downstaging protocols usually require an observation period (3 months in the study of Yao et al. [12]) with disease stability prior to waitlist registration. This period allows for the exclusion of tumors with high biological aggressiveness and an unacceptable risk of post-transplant recurrence. The Milan criteria are often used as the endpoint of downstaging protocols while the upper limits of tumor burden suitable for downstaging remain controversial [14].

The introduction of morphologic criteria resulted in a marked improvement in LT outcomes for HCC compared with the time before the introduction of the Milan criteria [15,16,17]. The effectiveness of these criteria in predicting post-transplant survival and ensuring an acceptable disease recurrence rate is based on the principle of “tumor burden”. Morphological characteristics of the tumor (size and number of the nodules) are considered a surrogate marker for the presence of microvascular invasion and/or poor differentiation of HCC [18,19], which are both independent predictors for HCC recurrence [20].

Since Mazzaferro’s original criteria, macroscopic vascular invasion has been considered an unacceptable risk factor for post-transplant recurrence. Over the years, how microvascular invasion is also associated with a high rate of recurrence has come to be understood [5,7].

Encouraging post-LT results have recently been reported for patients with complete regression of macroscopic vascular invasion after downstaging protocols including chemoembolization, radioembolization, or surgery. Indeed, recurrence rates are found to be acceptable (11%) in subjects presenting low alpha-fetoprotein (AFP) (<10 ng/mL) and complete vascular invasion regression at pre-transplant imaging. However, prospective validation is needed, and caution is required in considering this category of patients for transplantation [21].

The diagnosis of HCC in cirrhotic individuals is often based on radiological criteria in accordance with current guidelines, and liver biopsy is reserved for cases with nontypical imaging [22,23] so information about microvascular invasion is often lacking before transplantation. Although not routinely performed, histologic analysis can provide direct insight into tumor grading. Already in 2004, Cillo et al. reported that patients with moderately to well-differentiated HCC at pre-LT biopsy had a favorable 5-year survival rate of 75% and a recurrence-free survival rate of 92% [24]

More recently, the Toronto extended criteria have been proposed, which also consider preoperative histology. Using these criteria, liver transplantation could be offered to subjects with any number and size of hepatic nodules if they have no extrahepatic localization (including macrovascular invasion), no poorly differentiated neoplasm at preoperative histology of the larger nodule, and no cancer-related symptoms [25].

Serum alpha-fetoprotein (AFP) can also be considered a surrogate marker of HCC differentiation and vascular invasion [26], and the measurement of AFP serum levels before LT is a useful tool for identifying patients with a high risk of HCC recurrence [27], especially when associated with morphological criteria [28].

In 2018, Mazzaferro et Al. proposed the so-called Metroticket 2.0 model, which was constructed to predict post-transplant HCC-related specific death by considering AFP as a surrogate marker of microvascular invasion and morphological characteristics of HCC [29].

Also, in Toronto cohorts, high AFP levels at listing and at the time of transplant were associated with worse post-transplant outcomes, both in “Milan in” subjects and in subjects transplanted according to the extended Toronto criteria. AFP levels and trends over time appeared as relevant prognostic indicators, and a cut-off of 500 ng/mL was proposed as indicative of a poor prognosis [25].

In 2012, Duvoux et al. proposed and validated a model incorporating AFP, tumor size, and nodule number to predict post-transplant HCC recurrence: a cut-off of 2 was identified for the selection of patients with an acceptable risk of recurrence. In patients exceeding Milan criteria, a score ≤ 2 identifies subjects with low AFP values (<100 ng/mL) with an acceptable risk of 5-year post-transplant recurrence (14.4 ± 5.3%). On the contrary, in subjects within Milan criteria, a score higher than 2 identifies subjects with high AFP values (>1000 ng/mL) with an unacceptable risk of 5-year post-transplant recurrence (37.1 ± 8.9%) [30] (Table 2). 

### 2.1. Post-LT HCC Recurrence

Post-LT HCC recurrence is among the leading causes of death in subjects transplanted for this indication. Emerging data and expert consensus support post-transplant HCC surveillance, as early diagnosis and aggressive treatment have been demonstrated to improve survival outcomes [38,39,40].

Time to recurrence can be highly variable. HCC recurrence is an infrequent cause of death in the first year after LT (5.3% of deaths in the first year) [41]. A peak in HCC recurrence within 2–3 years after transplant has been identified by several studies, whereas recurrence after 5 years is less frequent [41,42,43,44]. Time to recurrence also represents a prognostic factor: early HCC recurrence is associated with a more severe prognosis [42,45,46].

Considering the very high impact of HCC recurrence on post-transplant prognosis, many of the criteria and scores for selecting transplant candidates were constructed with both post-transplant survival and HCC recurrence as outcomes. As previously mentioned, the HCC recurrence rate is largely dependent on the characteristics of the pre-transplant tumor. Using Milan criteria, recurrence has been observed in 5.7%–16% of cases, and this has become the benchmark for all the subsequent proposed criteria [46,47,48].

Data coming from analysis of the Organ Procurement and Transplantation Network/United Network for Organ Sharing (OPTN/UNOS) registry confirmed that some pre-transplant factors are strong predictors of recurrence, including the presence of extrahepatic or lymph nodal spread (OR 6.8 and 1.9, respectively), poor tumor differentiation (OR 2.8), micro- and macrovascular invasion (OR 2.6 and 3.2, respectively), explant TNM stage T4 or T3 (OR 2.4 and 1.9, respectively), downstaging from a stage > than T2 (OR 1.8), and high AFP values (OR 1.2) [48].

In a systematic review published in 2015, de’ Angelis et al. evaluated 61 heterogeneous studies. The observed overall mean recurrence rate was 16%, and the median time from LT to recurrence was 13 months (range 2–132 months). Most observed recurrences (67%) were extrahepatic. Interestingly, 51% of patients included in the study were transplanted beyond Milan criteria according to explant pathology [47]. Also, in a 30-year series by the University of California, Los Angeles (UCLA), recurrence was mainly extrahepatic, with liver allograft recurrence accounting for 37.8% and the pattern of recurrence largely multinodular [46].

In a recently published systematic review and meta-analysis of 125 studies including 55.333 patients, the pooled reported recurrence rate was 17% (n = 55,333, 95% CI: 15–19; I2 = 98.1%, *p* ≤ 0.001, τ2 = 0.01) with a non-statistically significant downward trend over the years. Geographical area and ethnicity have been shown to significantly influence the recurrence rate, ranging between 10% in South American studies and 21% in Asian studies, with Asian and African American men showing higher recurrence rates compared with Caucasians [49].

Tumor histology can provide useful information in predicting the risk of recurrence. Lasagni et al. analyzed a group of 70 patients with HCC recurrence and compared them with a matched cohort of patients without recurrence and validated their observations in an independent cohort. Patients experiencing HCC recurrence showed significantly increased expression of angiopoietin-2 in tumor endothelium but not in the hepatocyte at explant pathology. Univariate analysis identified BMI, log AFP at transplant, endothelial angiopoietin-2 expression, Milan criteria, Metroticket AFP score, and the AFP model as significantly associated with recurrence, while in multivariate analysis angiopoietin-2 expression was the only independent factor associated with recurrence (HR: 5.634, 95% CI 2.597–12.294, *p* < 0.001) [50].

Recently, six histological features of explant tissues were examined in a database of 380 patients undergoing LT for HCC: tumor region, normal liver tissue, portal area, fibrous tissue, hemorrhagic and necrotic tissue, and immune cells. Using artificial intelligence techniques with neural networks, all of these features, except for normal liver tissue, were used to create a Deep Pathomics Score (DPS) with the aim of predicting the risk of recurrence. DPS was able to discriminate between subjects with a low risk of recurrence (DPS ≤ 0.5161266, 5-year recurrence rates of 4.59%) and high risk of recurrence (DPS > 0.5161266, 5-year recurrence rates of 47.20%). Among the evaluated histological aspects, immune cells were the strongest prognostic determinant of recurrence in multivariate analysis (HR 1.907, 95% CI 1.490–2.440) [51]. Evidence exists that individuals transplanted for HCC have a distinct immune profile that is different from patients transplanted for other indications, and the balance of immunosuppression may influence the risk of graft rejection on the one hand and the risk of HCC recurrence on the other [52]. This was analyzed in detail in a recently published paper by Wei X. et al. [53]. In the immediate post-transplant period, a substantial but transient increase in myeloid and B cells and a decrease in T cells and natural killers was observed, though the populations of these cells returned to pre-transplant levels in 3 weeks. Plasmacytoid dendritic cells progressively increase after LT and peak 2 weeks after LT. Patients with and without HCC show very different immune cell composition, but there is only a small difference at this level in the first post-transplant period between HCC transplant recipients who will develop recurrence and patients who will not. Interestingly, a large amount of the differences in immune subsets between patients with and without HCC are evident even before transplantation. A more in-depth analysis showed that, starting from the third-week post transplantation, subjects who will develop HCC recurrence have a significant expansion of T14 (a subtype of γδ T lymphocytes) and T21 subpopulations (a subtype of CD8+ T cells) compared with patients who will not develop HCC recurrence [53]. On this basis, the identification of an “immune milieu” favoring recurrence would be conceivable.

As previously reported, AFP levels at the time of transplant are a strong predictor of HCC recurrence, but this remains a suboptimal biomarker. Des-gamma-carboxyprothrombin (DCP) and AFP bound to Lens culinaris agglutinin (AFP-L3) have been recently proposed as promising biomarkers. In a prospective cohort of 285 patients transplanted according to Milan criteria, AFP-L3 and DCP resulted in the strongest predictors of HCC recurrence, and a combination of both (AFP-L3 ≥ 15% and DCP ≥ 7.5) predicted 61.1% of observed recurrence [54] (Figure 1).

### 2.2. Post-LT HCC Surveillance

Evidence-based recommendations for HCC post-LT surveillance protocols are lacking to date as there are no adequately powered studies evaluating the best frequency, duration, and effects on LT outcomes [42,55], even if evidence exists that close surveillance is associated with better prognosis and greater feasibility of curative treatments [39].

Despite the absence of solid scientific evidence supporting the cost-effectiveness of post-transplant surveillance for HCC recurrence, there is broad agreement that periodic contrast imaging with CT or MRI should be performed in the post-LT setting and that periodic AFP should be obtained, particularly in subjects with high AFP values before LT. In subjects undergoing liver resection for HCC, current guidelines suggest indefinite cross-sectional imaging examinations of the abdomen and chest plus serum AFP every 3–6 months [56]. In the case of transplanted subjects, recurrence is determined, at least in part, by distinct mechanisms, and the risk is not directly comparable with liver resection in cirrhosis. In the absence of specific guidelines, we believe that an AFP-associated imaging evaluation should be performed for transplanted subjects every 3–6 months and for not less than 5 years post transplantation, although longer surveillance may be reasonable.

In this perspective, explant pathology can be useful in providing an estimate of the risk of HCC recurrence and therefore guide post-transplant surveillance intervals, reducing unnecessary radiation exposure. This is also crucial as suboptimal concordance data between pre-transplant imaging staging and pathological findings have been reported over time (only 44% of concordance reported by Freeman et al. in the OPTN/UNOS database, with 20% of patients without any evidence of tumor in the explant with no pre-transplant treatment) [57], although the better definition of imaging protocols [58] has more recently significantly increased concordance [48]. Explant-based prognostic models can provide a more realistic estimate of recurrence risk and allow specific surveillance and management protocols to be set up [59].

In 2017, the RETREAT score (Risk Estimation of Tumor Recurrence After Transplant) was proposed to stratify the risk of HCC recurrence in subjects transplanted with HCC meeting Milan criteria at pre-LT imaging. The 1061 patients included in the study were divided into a development cohort (721 subjects) and a validation cohort (341 subjects). A score including the presence of microvascular invasion at explant (present vs. absent), the sum of the largest diameter of viable tumors, the number of viable tumors at explant, and AFP levels at transplant was constructed in the development cohort and validated in the validation cohort. RETREAT score ranges from 0 to 8 and can stratify the risk of recurrence between very low risk (RETREAT score 0, 5-year recurrence risk of 2.9%) to extremely high risk (RETREAT score ≥ 5, 5-year recurrence risk of 75.2%) [60]. The score has been subsequently validated in a wider population from the UNOS database [36] and represents a reliable model to guide surveillance in HCC patients undergoing LT.

More recently, a new score based on the number of nodules, the size of the largest nodule, the presence of microvascular invasion, nuclear grade, and the last pre-LT AFP value—namely the Recurrence Risk Reassessment (R3)-AFP score—was tested and validated with a large multicentric cohort. This score ranges between 0 and 10 and stratifies patients into categories of very low risk of recurrence (R3-AFP score = 0 points, Recurrence risk 5.5%; 95% CI 3.5–8.7), low risk (R3-AFP score = 1–2 points, recurrence risk of 15. 1%; 95% CI 11.3–20.1), high risk (R3-AFP score = 3–6 points, recurrence risk of 39.1%; 95% CI 32.4–46.7), and very high risk (R3-AFP score > 6 points, recurrence risk of 73.9%; 95% CI 59.7–86.3). This score showed superimposable performance with the RETREAT score in both the test and the validation cohort [61]. This score, unlike the RETREAT score, was derived and tested on a population of patients that were transplanted also beyond the Milan criteria in some cases and is therefore probably more applicable in clinical practice considering the increasing use of extended criteria. A new score including maximum pre-transplant AFP values, immediate pre-LT neutrophil–lymphocyte ratio, explant pathologic variables such as the degree of vascular invasion, histological differentiation, and tumor size, has been recently proposed and validated in an American/European collaborative study using machine learning models. The RELAPSE score showed a good prediction of HCC recurrence (c-stat 0.78) in a large multicenter UNOS database including patients from different regions and with different selection criteria and showed similar performance in a European validation cohort (AUC 0.74-0.77) [62].

### 2.3. Effect of Immunosuppressive Therapy on Post-LT HCC Recurrence

The mainstay of post-LT immunosuppression in the early post-transplant period is the use of calcineurin inhibitors (CNIs) such as cyclosporine and tacrolimus. However, these drugs affect the immune system’s response and cancer immunosurveillance, thus promoting cancer cell survival and increasing the risk of HCC recurrence in a dose-dependent manner [63,64,65,66].

Mammalian target of rapamycin inhibitors (mTORi), such as sirolimus and everolimus, exhibit anti-tumoral growth effects (anti-proliferation and anti-angiogenetic) [67,68,69,70], and the mTORi-based immunosuppressive regimen shows decreased risk for post-transplant HCC recurrence compared to CNIs [71,72,73,74,75,76,77,78]. In the SiLVER trial, an international multicentre randomized control trial investigating sirolimus-based regimens versus mTORi-free regimens in LT for HCC, no significant differences were found in long-term overall or recurrence-free survival between sirolimus-treated patients and patients not taking mTORi. An initial benefit was observed in the first 5 years post-transplant, especially in subjects at lower risk of recurrence with tumors within the Milan criteria, though this was not maintained over time [79].

On this basis, the current recommendation is to taper immunosuppression to the lowest effective dose for protection against graft rejection and reconsider immunosuppressive therapy regimens by combining or switching to an mTOR and decreasing the dosage of CNIs [80].

### 2.4. Treatment of Post-LT HCC Recurrence

Patients with post-LT HCC recurrence should be promptly treated. As in the pre-transplant setting, curative treatments when feasible provide the best results in terms of long-term survival [40,43,46,81,82,83].

In 2015, Sapisochin et al. proposed a prognostic score for HCC recurrence using three variables independently associated with poor prognosis in a multicentre cohort of patients with HCC recurrence. The three variables considered were the following: being non-amenable to curative-intent treatments (surgery or ablation), early recurrence (<1 year), and AFP values of ≥100 ng/mL at the time of recurrence. Patients with a 0 score (none of the considered variables) showed 5-year survival of ∼50% [81]. The score was subsequently validated in an independent multicentre set of patients, confirming its predictive ability even though AFP values were not confirmed to be significant predictors of poor prognosis [84]. Interestingly, in both studies, the number of recurrent nodules was not significantly associated with prognosis even though candidacy for therapies with curative intent can be considered a surrogate for tumor burden.

In many of the papers that have analyzed the treatment of post-transplant recurrence, surgical treatments (either with open or minimally invasive surgery) and percutaneous or laparoscopic ablative treatments are considered together, both because of the small sample size and because, in the case of small nodules (up to 2–3 cm in maximum diameter), the results of the two approaches are comparable.

In the UCLA series, the majority of patients experiencing post-LT HCC recurrence (59.9%) received non-surgical therapy, 23.3% received surgery (alone or in combination with non-surgical treatments), and 18% received only supportive treatments. Surgical treatments achieved the best results in terms of survival, especially in the case of graft surgical resection for intrahepatic oligo-recurrence (median survival of 27.8 months). In this study, the patients were followed with a surveillance protocol including serum alpha AFP and axial contrast-enhanced imaging every 6 months post LT. Despite this intensive surveillance program, HCC recurrence was often not suitable for curative treatments because of the advanced stage or because of technical issues. Interestingly, the authors found that the pattern of tumor recurrence (time to recurrence, number, size, and location of recurrent nodules), rather than explant pathology, was the main factor affecting post-recurrence survival [46]. Similarly, both de’ Angelis et al. [47] and Kornberg et al. [43] reported higher survival in patients with post-LT recurrence suitable for surgical treatments (median survival was 42 months and 65 months, respectively). Finally, an Italian multicentric study showed a better 4-year survival rate of 57% in patients undergoing surgical resection for intra- and extrahepatic HCC recurrence compared to the 14% that was observed for unresectable disease [83].

Extrahepatic recurrence should not be a reason for exclusion from surgical treatments, as various studies have shown a survival advantage in subjects undergoing resection for such recurrences [85,86,87,88].

As in the previously cited series, Fernandez-Sevilla et al. reported significantly longer survival in patients undergoing surgical resection compared to that of non-resected patients (35 vs. 15 months). Most resected patients received surgery for extrahepatic sites (2 intrahepatic vs. 20 extrahepatic), and patients with extrahepatic unilocular recurrence presented the best results after resection. In contrast, the intrahepatic location of the recurrence was found to be significantly associated with worse outcomes (HR 1.8), together with AFP level > 100 ng/mL at relapse (HR 2.01) and multifocal recurrence (HR, 1.79) [82].

Among extrahepatic recurrences, multiple experiences are reported in the literature about pulmonary recurrences with satisfactory outcomes for treated patients [89,90,91,92,93].

However, it must be emphasized that the surgical treatment of post-LT recurrent HCC poses some issues to be kept in mind. First of all, transplanted subjects are at higher risk of peri-operative infective complications due to immunosuppression. Furthermore, as concerns the resection of hepatic or perihepatic recurrences, the possible occurrence of post-surgical adhesions should be considered, which can be particularly challenging at the hilar site and, in general, near surgical anastomoses. In recent years, liver resections using minimally invasive surgical techniques have been increasingly performed, both for primary and recurrent HCC [94]. The use of minimally invasive surgery has been associated with a reduction in blood loss and operating time [95,96,97]. With increasing surgical experience in minimally invasive surgery, this approach has also been used in the treatment of post-LT HCC recurrence, with encouraging results in highly selected cases [98,99]. Finally, in the case of both hepatic and extrahepatic resections, the prerequisite of good graft function is essential [100,101,102].

In patients who are not candidates for surgical resection or in the case of small recurrent lesions, locoregional techniques are viable alternatives. Among these, ablation has shown the best results. Ablation can be performed with two major techniques: radiofrequency ablation (RFA) and microwave ablation (MWA).

Percutaneous or laparoscopic RFA is ideal for small tumors deeply located in the liver parenchyma and with limited anatomical relationships to vessels and other abdominal organs. In general, it is well known that RFA provides equivalent local disease control compared with surgical resection for nodules up to 3 cm in diameter. In these patients with initial intrahepatic post-LT recurrence, the less invasive nature of such procedures is an additional advantage over surgery due to lower morbidity and the lower risk of complications, although comparison studies in this setting are limited [103,104,105].

In an Asian cohort of patients with post-transplant HCC recurrence, the authors found no significant differences in the 5-year overall survival of patients treated with RFA or surgical resection, but not statistically significant worse disease-free survival was observed in RFA-treated subjects (5-year OS 28% vs. 35%; DFS 0% vs. 16%). It should be noted that in the treatment algorithm used in this study, RFA was used exclusively in cases of contraindication to surgery, and given the small number of patients, balancing for site and type of recurrence could not be performed [106].

Although numerically limited, there are published experiences in the literature about the use of the MWA ablation technique, either percutaneously or laparoscopically, for post-transplant recurrences, with some of these experiences describing encouraging results [107,108].

In a small retrospective series of 11 patients with post-LT recurrence, MWA ablation was well tolerated and only 3 of 11 patients exhibited tumor progression after treatment; the 2-year survival rate was 15.3% and the mean survival time was 17.3 months [107].

No specific studies have evaluated this technique in post-LT recurrence in comparison with RFA or surgery.

Trans-arterial chemoembolization (TACE) is widely used in pre-transplant settings both as a bridge treatment and in downstaging protocols. In the case of post-LT recurrence, it is feasible, but results seem to be worse than those achieved using surgery or ablation.

Ko et al. reported no complications in a group of 28 patients treated with one or more cycles of TACE for recurrent HCC after living donor liver transplantation. A large percentage of treated subjects experienced a reduction in target tumor volume of 25% or more; however, 92.9% of patients experienced intra- or extrahepatic recurrence in the first 6 months after the procedure. The 1-, 3-, and 5-year survival rates after TACE were 47.9%, 6.0%, and 0%, respectively. It should be noted, however, that the enrolled population is numerically limited and very heterogeneous, with some patients having largely multinodular recurrence [109].

Recently, better outcomes have been reported in an Asian retrospective cohort. In a group of 54 patients treated with TACE, the 1-, 2-, and 3-year overall survival rates were 44.71%, 42.2%, and 19.5%, respectively. Significantly better results were observed in a group of 52 propensity score-matched patients treated with TACE plus Sorafenib (1-, 2-, and 3-year overall survival of 68.9%, 48.4%, and 35.2%, *p* = 0.035) [110].

Better results have been reported by Zhou et al., who observed 1-, 2-,3-, and 4-year overall survival rates after OLT that were 85.7%, 56.3%, 37.6%, and 18.8%, respectively [111]. In this study, the procedure was also well tolerated with transient adverse effects in the immediate post-procedure phases that resolved within a few days.

Systemic therapies, mainly tyrosine kinase inhibitors (TKIs), have been extensively studied in non-transplanted patients with advanced HCC [112,113].

In 2013, a retrospective case–control Italian study explored the use of Sorafenib in patients with post-LT HCC recurrence not eligible for surgery or locoregional techniques. Sorafenib provided better outcomes compared with the best supportive care, with post-recurrence survival of 21.3 months in treated subjects compared to 11.8 months in non-treated subjects [114]. Sorafenib side effects, particularly if combined with immunosuppressive drugs, could limit its use.

The association of Sorafenib and mTORi has been evaluated in a retrospective study including 31 patients. Median overall survival after the initiation of mTORi + Sorafenib was 19.3 months. The time to disease progression after treatment initiation was 6.77 months. In this cohort, 14 patients presented a treatment response (1 patient with a partial response and 13 with stable disease during the follow-up) [115].

Recently, the safety of early Sorafenib administration for post-LT HCC recurrence was evaluated in a retrospective cohort including 50 patients. All the enrolled patients presented a Sorafenib-related adverse event, with 56% of them being grade 3–4. There was no significant difference in Sorafenib-related adverse events between patients taking mTORi or other immunosuppressive regimens, except for any grade of hand–foot syndrome. Interestingly the use of mTORi + Sorafenib was associated with an increase in overall survival in univariate analysis but not in multivariate analysis [116].

Lenvatinib has also recently been used for the treatment of post-transplant HCC recurrence.

A recent multicentre international retrospective study evaluating 45 Lenvatinib-treated subjects with post-LT recurrent HCC reported median progression-free survival of 7.6 months (95% CI: 5.3–9.8 months) and overall survival of 14.5 months (95% CI: 0.8–28.2 months) over a 12.9-month median follow-up. In total, 20% of patients presented a partial response, 68% stable disease, and 6.7% progressive disease during treatment. These results are comparable to those found in non-transplanted subjects. No effect on survival was found for the immunosuppressive regimen in this study. The majority of analyzed patients experienced an adverse event (97.8%) but the overall safety profile of the drug appeared satisfactory, with only 8.9% of subjects discontinuing the drug due to adverse events [117].

Yang et al. analyzed a retrospective cohort of patients with post-LT recurrent HCC treated with various approaches. In the sub-group of patients treated with TKIs progressing during Sorafenib treatment, Sorafenib-tolerant subjects receiving Lenvatinib showed significantly longer survival when compared with patients shifted to Regorafenib or patients just stopping Sorafenib. Also, in this report, the use of mTORi was associated with better overall survival if compared with CNIs [118].

Second-line systemic therapy has been evaluated in the post-LT setting. In two multicentre international studies, Iavarone et al. evaluated the efficacy and the safety of Regorafenib in patients with post-LT HCC recurrence tolerant to Sorafenib but progressing during treatment. In the first study, the safety of Regorafenib was evaluated in 28 subjects. All enrolled subjects presented at least one adverse event, with fatigue, weight loss, and dermatological reactions the most frequently observed (75%, 54%, and 50%, respectively). Severe adverse events occurred in 43% of subjects. During treatment, immunosuppressive drug adjustment was needed in five patients because of increased circulating levels of the immunosuppressive drugs. The median OS was 12.9 months with a median follow-up of 8.9 months (1.5–24.0) [119]. In the second study, the authors evaluated the efficacy of Regorafenib after Sorafenib discontinuation (36 patients) compared with best supportive care (45 patients). Regorafenib-treated subjects showed significantly longer overall survival (13.1 months vs. 5.5 months, *p*: <0.01) and the treatment was well tolerated, with 44% of patients receiving the full dose of the drug [120] (Figure 2).

Recently the introduction of immunotherapy with immune checkpoint inhibitors (ICIs) has changed the scenario of systemic treatment for advanced HCC in non-transplanted patients [121,122]. The use of such drugs in the post-transplant setting to treat HCC recurrence or other post-transplant cancers poses relevant doubts about safety considering the risk of immune-mediated adverse events that, in the post-transplant period, may include rejection, graft loss, and even death [123,124].

Data on the efficacy and safety of ICIs for the treatment of HCC recurrence in LT recipients are largely lacking, with the few existing examples in the literature represented by case reports, small case series, and single-centre experiences with contrasting results ranging from optimal responses to catastrophic events [123,125,126,127,128].

The time elapsed between liver transplantation and the administration of ICIs seems to have an influence on the development of liver allograft rejection, with higher risk in the early post-transplant period. The minor risk of adverse events in long-term transplanted patients could be due to transplant immunological tolerance and reduced immunosuppression needs [129,130,131,132,133].

Further and adequately designed studies are needed to better understand the efficacy and safety of these treatments in transplant recipients in general and in post-LT HCC recurrence in particular.

## 3. Conclusions

Post-LT HCC recurrence dramatically worsens the prognosis of transplanted patients. Several pre-transplant factors have been identified as risk factors for recurrence, and specifically designed scores are available to predict the risk of recurrence and guide post-transplant surveillance and management. Immunosuppressive protocols may affect the risk of HCC recurrence post-LT but the best immunosuppressive schedule for patients transplanted for HCC is yet to be defined. Among treatments for HCC recurrence, surgery, whether for hepatic or extrahepatic recurrence, shows the best results and should be the preferred approach of treatment in selected patients with a limited burden of tumor recurrence. In more advanced cases, systemic therapy with TKIs demonstrates promising results in terms of improved patient survival.

### Future Directions

LT for HCC is a growing indication. In this context, the introduction of new biomarkers to clinical practice is needed to provide a deeper understanding of tumor biology, to allow for better patient selection, and to stratify post-LT recurrence risk.

Prospective validation of predictive models is needed to implement customized surveillance programs, and specifically designed studies are needed to evaluate the cost-effectiveness of post-transplant surveillance protocols.

The definition of personalized immunosuppressive schemes and drug levels with the measurement of specific biomarkers of immune system activity could help in reducing the risk of HCC recurrence and improve the management of recurrent cancer.

The potential use of systemic adjuvant treatments and the use of ICIs could significantly change the prognosis and management of HCC recurrence post LT in the near future.

## Figures and Tables

**Figure 1 cancers-15-05593-f001:**
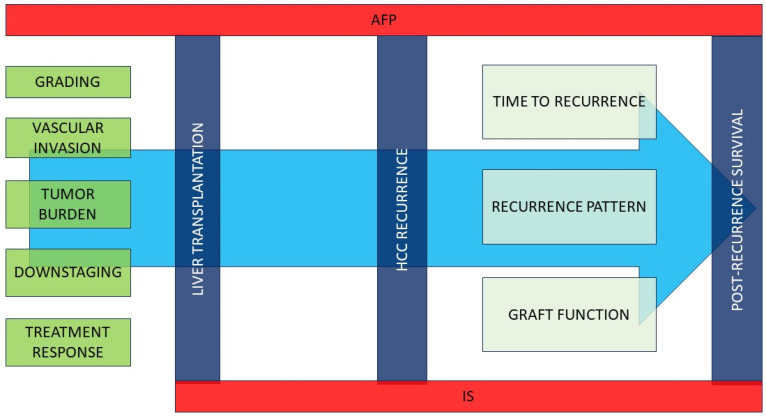
Pre- and post-transplant factors affecting HCC recurrence and post-recurrence survival.Various pre-transplant characteristics may affect the risk of post-transplant HCC recurrence. AFP is a powerful predictor of both recurrence risk and post-recurrence mortality. The immunosuppressive regimen may affect both the risk of neoplasm recurrence and post-recurrence survival. Once recurrence occurs, time to recurrence, pattern of recurrence (and consequently applicable treatments), and residual graft function are the main predictors of mortality. IS: immunosuppression; VI: vascular invasion; AFP alpha-fetoprotein.

**Figure 2 cancers-15-05593-f002:**
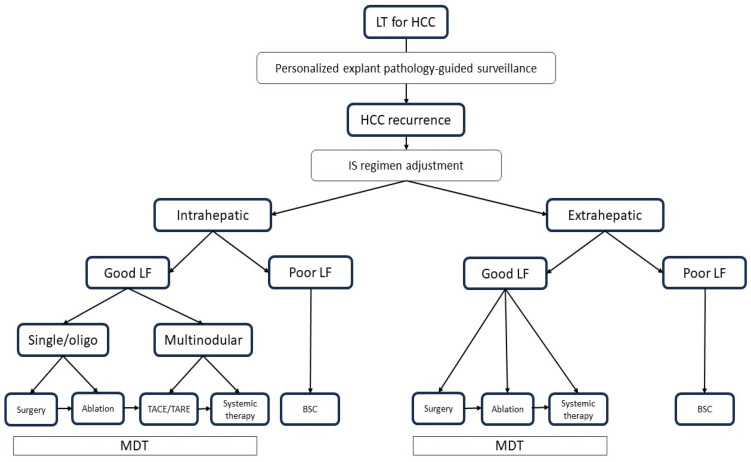
Proposed treatment algorithm for HCC recurrence. Recurrence pattern and liver function have a major influence on treatment. Whenever possible, through discussion in a multidisciplinary team, the treatment with the best expected results should be offered, only switching to treatments of lower efficacy if more radical treatments are not applicable. IS: immunosuppression; LT: liver transplantation, HCC: hepatocellular carcinoma, IS: immunosuppression, LF: liver function; TACE: trans-arterial chemoembolization, TARE: trans-arterial radioembolization, BSC: best supportive care; MDT: multidisciplinary team.

**Table 1 cancers-15-05593-t001:** Selection criteria for liver transplantation for hepatocellular carcinoma not considering AFP.

Model Name	Population	Primary Outcome	Performance
Milan criteria (1996) [2]	Adult HCC patients that underwent LT	Overall survival	5-year OS: 85% 5-year RFS: 92%
UCSF (2001) [4]	Adult HCC patients that underwent LT	Overall survival	5-year OS: 75.2% 5-year RFS: 80.9%
Up-to-7 criteria (2009) [5]	Adult HCC patients that underwent LT	Overall survival	5-year OS: 71.2% (Beyond MC and within up-to-7 criteria)

HCC: hepatocellular carcinoma; LT: liver transplantation; OS: overall survival; RFS: recurrence-free survival; MC: Milan criteria, UCSF: University of California, San Francisco.

**Table 2 cancers-15-05593-t002:** Selection criteria for liver transplantation for hepatocellular carcinoma including AFP.

Model Name	Population	Primary Outcome	AFP Details	Performance
Seoul criteria (2007) [31]	Adult HCC patients who underwent LDLT	Overall survival	Last AFP ≤ 20, 20.1–200, 200.1–1000, >1000 ng/mL	3-year RFS: score 3–6, 87%; score 7–12, 31% 3-year OS: score 3–6, 79%; score 7–12, 38%
AFP model (2012) [30]	Adult HCC patients diagnosed before listing that underwent primary LT	HCC recurrence	Log_10_(AFP_L_) Simplified version: low-risk AFP ≤1000 or 100–1000 ng/mL high-risk AFP > 1000 ng/mL	5-year recurrence: score ≤ 2, 8.8%; score > 2, 50.6%
AFP-TTD criteria (2012) [32]	Adult HCC patients who underwent LT	HCC recurrence	Last AFP ≤ 400 ng/mL	Recurrence rate (median FU 43 months): in criteria, 4.9%; outside criteria 33.0%
TTV/AFP model (2012) [33]	Adult HCC patients that underwent LT (beyond MC, within TTV/AFP)	HCC recurrence	AFP≤ 400 ng/mL	In MC and in TTV/AFP: 4-year DFS: 77.9%; 4-year OS: 78.7% Out MC and in TTV/AFP: 4-year DFS: 68.0%; 4-year OS: 74.6%
TRAIN score (2016) [34]	Adult HCC patients who received locoregional therapy before LT	HCC recurrence	AFP slope ≥ 15 ng/mL/months	(Validation set) ITT 5-year survival in/outside criteria: 66.7%/20.7% ITT 5-year recurrence rate in/outside criteria: 13.8%/100%
Extended Toronto criteria (2016) [25]	Adult HCC patients diagnosed before listing that underwent LT	Overall survival	AFP_L_ < 500 ng/mL	1-, 3-, 5-year patient survival (beyond MC, within ETC): <500 ng/mL 60%, 43%, 37% ≥ 500 ng/mL 88%, 73%, 64%
Pre-MORAL (2017) [35]	Adult HCC patients who underwent LT	HCC recurrence	Maximum AFP from HCC diagnosis to LT ≥ 200 ng/mL	5-year RFS: low risk (score 0–2): 98.6% medium risk (score 3–6): 69.8% high risk (score 7–10): 55.8% very high risk (score > 10): 0% (1-year RFS 17.9%)
RETREAT (2017) [36]	Adult HCC patients, preoperatively always within MC, with MELD exception that underwent LT	HCC recurrence	Pre-operative AFP: 0–20 ng/mL; 21–99 ng/mL; 100–999 ng/mL; ≥1000 ng/mL	3-year recurrence risk: score 0, 1.6%; score 1, 5%; score 2, 5.6%; score 3, 8.4%; score 4, 20.3%; score ≥5, 29.0%
NYCA (2018) [37]	Adult HCC patients that underwent LT	HCC recurrence	AFP response (max AFP to final AFP)	5-year RFS: low risk (score 0–2): 90% acceptable risk (score 3–6): 70% high risk (score ≥7): 42%
Metroticket 2.0 (2018) [29]	Adult HCC patients that underwent DBD LT	HCC-specific survival	Pre-transplant AFP: <200, 200–400 ng/mL, 400–1000 ng/mL, >1000 ng/mL	5-year RFS: within criteria 89.6% vs. beyond criteria 46.8% 5-year OS: within criteria 79.7% vs. beyond criteria 51.2% (tumor-specific survival 93.5% within vs. 55.6% beyond)

HCC: hepatocellular carcinoma; AFP: alpha-fetoprotein; LDLT: living donor liver transplantation; RFS: recurrence-free survival; LT: liver transplantation; TTD: total tumor diameter; FU: follow-up; TTV: total tumor volume; TRAIN: Time-Radiological-response-Alpha-fetoprotein-Inflammation score; ITT: intention to treat; MC: Milan criteria; ETC: extended Toronto criteria; MORAL: Model Of Recurrence After Liver Transplant; RETREAT: Risk Estimation of Tumor Recurrence After Transplant; NYCA: New York–California; DBD: donation after brain death.

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
