# Peer review of "Therapeutic Approach to Post-Transplant Recurrence of Hepatocellular Carcinoma: Certainties and Open Issues"

_cancers, 2023, doi:10.3390/cancers15235593_

Round 1
Reviewer 1 Report
Comments and Suggestions for Authors
The manuscript is well written and present a 360° terapeutic option for the treatment of recurrent HCC after LT.
I have a minor suggestion to the authors.
in the section lines -271-276- the authors didn't reported the possibility to treat with minimally invasive surgery redo resection for recurrence, even in patient after LT. (10.1007/s13304-021-01200-6). As correctly mentioned there was first a conviction to have technical issues with post operative adhesions. However, with the minimally approach more and more frequently redo surgery are performed without an increased operative morbidity.
In my opinion this point should be mentioned.
Author Response
Revierwer 1 comments:
in the section lines -271-276- the authors didn't reported the possibility to treat with minimally invasive surgery redo resection for recurrence, even in patient after LT. (10.1007/s13304-021-01200-6). As correctly mentioned there was first a conviction to have technical issues with post operative adhesions. However, with the minimally approach more and more frequently redo surgery are performed without an increased operative morbidity. In my opinion this point should be mentioned.
Author’s response: we have added the suggested reference and briefly mentioned the possibility of a minimally invasive surgical approach
Reviewer 2 Report
Comments and Suggestions for Authors
This manuscript titled “Therapeutic approach to post-transplant recurrence of hepatocellular carcinoma: certainties and open issues” is a review of HCC in transplant with a discussion of management post recurrence. This is an important topic for transplant oncology and does a nice brief overview, although I have some comments:
1. This is nicely written however lacks novelty as this has been written quite a few times.
2. There is a host of data out there about HCC post-transplant recurrences including scoring systems. The descriptions in each aspect of the manuscript are sparse and have been reviewed in the past in more detail. I would suggest more detail with regards to guiding the therapy post.
3. I would define surgical therapy for the paper. Some of the cited works have ablation in the same category.
4. There is a whole section on surveillance of HCC however no description of the actual recommendations.
5. From a writing style there are many 1-2 sentence paragraphs that could be combined.
6. The figure is not that helpful in the manuscript.
Comments on the Quality of English Languagesee above
Author Response
- This is nicely written however lacks novelty as this has been written quite a few times.
Author’s response: we agree with reviewer comment. However, we believe that the review of the topic may be of interest especially for recent insights into systemic therapies and the potential future use of immunotherapeutic drugs. In the “Post-LT HCC recurrence” section we also have included the discussion of some recently published papers on molecular, biological, and histological factors associated with recurrence.
- There is a host of data out there about HCC post-transplant recurrences including scoring systems. The descriptions in each aspect of the manuscript are sparse and have been reviewed in the past in more detail. I would suggest more detail with regards to guiding the therapy post.
Author’s response: We thank the reviewer for the suggestion. We have detailed further aspects of HCC recurrence and recalled that several scores have been proposed for predicting recurrence. We also reported some examples of prognostic indicators in HCC recurrence.
- I would define surgical therapy for the paper. Some of the cited works have ablation in the same category.
Author’s response: we have reported how surgical and ablative treatments are often considered together in studies that have analysed treatments for recurrent HCC.
- There is a whole section on surveillance of HCC however no description of the actual recommendations.
Author’s response: we have reported that there is no specific recommendations for surveillance of HCC recurrence in LT recipients but we have proposed a surveillance protocol by similarity with subjects undergoing liver resection with customisation based on the risk of recurrence estimated with specific scores
- From a writing style there are many 1-2 sentence paragraphs that could be combined.
Author’s response: we returned to the structure of the text by combining paragraphs that were too short
- The figure is not that helpful in the manuscript.
Author’s response: we think that the figure may increase the clarity of the text and we included also a new figure with a proposed treatment algorithm
Round 2
Reviewer 2 Report
Comments and Suggestions for Authors
The authors addressed many of my concerns and the other reviewer"s comment. I do worry that it lacks novelty but overall it is well written and does have some updates.